# Concerted dynamics of metallo-base pairs in an A/B-form helical transition

Olivia P. Schmidt[1], Simon Jurt[1], Silke Johannsen [1], Ashkan Karimi[1], Roland K.O. Sigel[1] & Nathan W. Luedtke[1]*

Metal-mediated base pairs expand the repertoire of nucleic acid structures and dynamics. Here we report solution structures and dynamics of duplex DNA containing two all-natural C-Hg$^{II}$-T metallo base pairs separated by six canonical base pairs. NMR experiments reveal a 3:1 ratio of well-resolved structures in dynamic equilibrium. The major species contains two (N3)T-Hg$^{II}$-(N3)C base pairs in a predominantly B-form helix. The minor species contains (N3)T-Hg$^{II}$-(N4)C base pairs and greater A-form characteristics. Ten-fold different $^{1}J$ coupling constants ($^{15}$N,$^{199}$Hg) are observed for (N3)C-Hg$^{II}$ (114 Hz) versus (N4)C-Hg$^{II}$ (1052 Hz) connectivities, reflecting differences in cytosine ionization and metal-bonding strengths. Dynamic interconversion between the two types of C-Hg$^{II}$-T base pairs are coupled to a global conformational exchange between the helices. These observations inspired the design of a repetitive DNA sequence capable of undergoing a global B-to-A-form helical transition upon adding Hg$^{II}$, demonstrating that C-Hg$^{II}$-T has unique switching potential in DNA-based materials and devices.

[1] Department of Chemistry, University of Zurich, Zurich, Switzerland. *email: nathan.luedtke@chem.uzh.ch

Transitions between A- and B-form duplexes were discovered by Franklin and Gosling when conducting X-ray fibre diffraction analyses under various humidities[1]. In solution, protein binding reactions can also partially dehydrate duplex DNA, giving global A-form viral genomes[2], as well as local A-form perturbations at specific binding sites[3]. Small molecules such as polyamines[4], aminoglycosides[5,6], hexaamminecobalt (III)[7], and cisplatin[8,9] can induce local B → A transitions via mechanisms independent of global changes in hydration and water activity. The binding of metal ions to discreet coordination sites[10] in nucleic acids can be coupled to the (re)folding of DNA and RNA molecules[11–16] that activate DNAzymes[17,18], ribozymes[19,20], riboswitches[21–23], and DNA-based materials[24–31]. Previous studies mostly focused on characterizing the changes in structure and function of metal-bound versus metal-free (apo) nucleic acids. Here we report the structures and dynamics of two interconverting structures of the same metallo duplex, where local changes in metal-nucleobase ionization and isomerization are directly coupled to a global conformational exchange. This observation, together with the structural differences between metal-bound and apo duplex structures, enabled our design and identification of a duplex DNA containing 15 C-T mismatches that exhibits a global B → A helical transition upon adding Hg[II].

Transition metal ions participate in reversible binding interactions between opposing nucleobases[32–37]. T-Hg[II]-T provided the first such example of an all-natural metal-mediated base pair composed of a pyrimidine-pyrimidine mismatch and a transition metal ion[38–40]. T-Hg[II]-T base pairs exhibit very high kinetic and thermodynamic stabilities[41–43], and can serve as functional mimics of T-A by causing enzymatic misincorporation of dTTP across from thymidine to give T-Hg[II]-T in vitro[44] and possibly in vivo[44,45]. NMR studies confirmed early predictions that Hg[II] binds to T-T mismatches via N3 coordination of two deprotonated thymidine residues[46]. Structurally analogous C-Ag[I]-C base pairs have also been reported[47], and in both these cases, little or no impact on the global structure of the B-form duplex was reported[47–49].

C-Hg[II]-T is a newly discovered, all-natural metallo base pair for which relatively little information is available[32,50,51]. Using fluorescent nucleobase analogues and [1]H NMR spectroscopy, we recently reported stoichiometric, high affinity binding of Hg[II] to DNA duplexes containing C-T mismatches[51]. Conducted in parallel, crystal screening of various oligonucleotides and metal ions produced an X-ray structure of a short (8-mer), A-form DNA sequence containing two C-Hg[II]-T base pairs with an unexpected metal binding mode involving the exocyclic amine (N4) of a deprotonated cytosine "C" residue and (N3) of thymine "T" (Fig. 1)[50]. This coordination mode was in contrast to a preliminary proposal for (N3)T-Hg[II]-(N3)C coordination based on structural homology with T-Hg[II]-T and small increases in thermal stabilities of duplexes containing C-T mismatches after

adding Hg[II] (Fig. 1)[32]. The global A-form structure observed in the crystal structure was inconsistent with circular dichroism (CD) data of slightly longer, 14–21-mer duplexes containing one or two C-Hg[II]-T base pairs[51]. The CD spectra suggested B-form helices, and little-to-no changes in their global conformation upon adding Hg[II]. The metal binding mode(s) and global structural characteristics of duplex DNA containing C-Hg[II]-T base pairs in solution were therefore unclear.

Here we report a detailed NMR study using [15]N-labelled DNA and [199]Hg enriched mercury salts to determine the solution structures and dynamics of C-Hg[II]-T base pairs in duplex DNA. Unlike previous examples of metal-mediated base pairs, C-Hg[II]-T exhibits two types of covalent connectivities that are dynamically coupled via a global conformational change in helical structure. A palindromic, 14-mer duplex with two C-Hg[II]-T sites separated by six canonical base pairs (ODN[1], Fig. 2a, Supplementary Table 1, and Supplementary Figs. 1, 2) exists as a 3:1 mixture of well-defined duplexes in dynamic equilibrium. Both structures exhibit groove and rise dimensions intermediate between ideal A- and B-form helices. The most abundant duplex contains (N3)T-Hg[II]-(N3)C connectivity and mostly B-form helical characteristics, whereas the minor species contains (N3)T-Hg[II]-(N4)C base pairs and more A-form characteristics. No indication of a third duplex containing one of each type of metallo base pair is evident, consistent with long-range conformational coupling between the two metal centres. Furthermore, the rate constants for nucleobase-metal-nucleobase isomerization ($k_{forward} = 3.5\ s^{-1}$ and $k_{reversed} = 7.7\ s^{-1}$) measured using [[15]N,[1]H]-HSQC experiments are nearly identical to those of the global conformational exchange of duplex structures measured using [[1]H,[1]H]-NOESY experiments ($k_{forward} = 4.3 \pm 0.6\ s^{-1}$, $k_{reversed} = 8.8 \pm 0.9\ s^{-1}$). These results therefore support the coupling of metal-ligand isomerization reactions over long distances (> 20 Å) via a global conformational change of the double helix. Taken together with the greater A-form characteristics upon metal binding, these results suggest that placing numerous C-T mismatches throughout a repetitive duplex sequence can facilitate a global B → A helical transition upon adding Hg[II]. To test this possibility, we prepare and analyse a small library of hairpin duplex DNAs (n = 10) and identify a duplex sequence that adopts a global A-form structure upon adding stoichiometric Hg[II]. This helical transition is rapid (< 30 s) and fully reversible upon addition of N-acetylcysteine in a cycle that can be repeated more than 10 times on

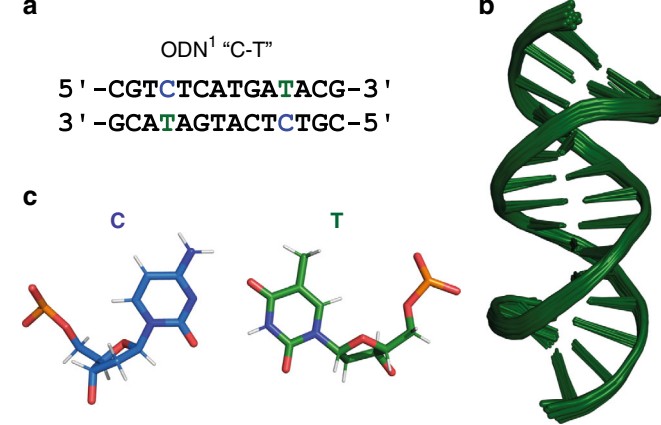

**a**

ODN[1] "C-T"

5'–CGT**C**TCATGA**T**ACG–3'
3'–GCA**T**AGTACT**C**TGC–5'

**b**

**c**

 C   T

**Fig. 2** Solution structure of metal-free duplex ODN[1] "C-T". **a** DNA sequence. Blue and green bases indicate C-T mismatches. **b** Overlay of 20 lowest energy structures. The structures were aligned to a representative model (#1) containing fewest outliers in the geometric quality criteria. **c** Facial recognition of non-planar, C-T pair in model #1 (PDB 6RLS)

**Fig. 1** Proposed C-Hg[II]-T binding modes. **a**, (N3)T-Hg[II]-(N3)C coordination based on structural homology with T-Hg[II]-T[32] and **b**, (N3)T-Hg[II]-(N4)C metal binding mode found in a short, A-form duplex DNA[50]

(N3)T-Hg[II]-(N3)C

(N3)T-Hg[II]-(N4)C

the same DNA. In addition to its broad implications in structural biology and biochemistry, such demand control of B → A and A → B helical switching may be utilized in the future development of advanced DNA-based materials and devices.

## Results

**Solution structure of Hg$^{II}$-free (apo) duplex ODN$^1$ "C-T".** Our preliminary $^1$H NMR and CD studies utilized a 14-mer $C_2$ symmetric, self-complementary sequence containing two C-T mismatches (ODN$^1$ "C-T", Fig. 2a)[51]. This sequence exhibits a thermodynamically predictable[52], two-state transition between single strand and duplex (Supplementary Fig. 2), as well as a $^1$H NMR exchangeable imino region that is well resolved at pH = 7 (Supplementary Fig. 3). Here the full assignment of proton resonances was conducted by sequential walking along H1′ and aromatic protons (H1′$_n$ → H6/H8$_{n+1}$ → H1′$_{n+1}$) (Supplementary Fig. 14). As cross references for our assignments, H2′/aromatic, H2″/aromatic (H2′/H2″n → H6/H8$_{n+1}$ → H2′/H2″$_{n+1}$), and aromatic/aromatic (H6/H8$_n$ → H6/H8$_{n+1}$ → H6/H8$_{n+2}$) regions in the [$^1$H,$^1$H]-NOESY, [$^1$H,$^1$H]-TOCSY (H1′ → H2′/H2″) and [$^{13}$C,$^1$H]-HSQC spectra (aliphatic and aromatic regions) were used. All signals of the duplex were well resolved and the sequential walk could be followed through the entire sequence. Models for the duplex were constructed based on 958 conformationally restrictive nuclear Overhauser effect (NOE) distance restraints (Table 1). The models did not contain any artificial constraints of co-planarity or hydrogen bonding for the C-T mismatches. Superimposition of the 20 lowest energy structures from 200 computed structures gave an overall root mean square deviation (r.m.s.d) of all heavy atoms of 0.74 ± 0.26 Å, and 0.54 ± 0.21 Å for the C-T mismatch (Table 1, Fig. 2b). The central region forms a canonical B-form duplex, yet local perturbations about the non-coplanar C-T mismatches cause a 19 ± 3° bend in the helical axis at each mismatch. Given the high dynamics of this system and limitations of the modelling used, it is difficult to ascertain the exact pattern(s) of hydrogen bonding present in C-T mismatches. Consistent with an early NMR model, the mismatched pyrimidines are stacked inside the duplex in a co-facial orientation (Fig. 2c)[53]. Our structure suggests C-T mismatches containing only one or two very weak hydrogen bonds. Evidence for weak hydrogen bonding is observed in the C-T imino proton resonances at 10.9 ppm that are broader than the imino resonances of

an analogous duplex ODN$^2$ "T-T" containing T-T mismatches, and much broader than those of ODN$^3$ "G-T" containing G-T wobble base pairs (Supplementary Fig. 3). These observations correlated very well with the thermal stabilities of these duplexes (T$_m$ ODN$^1$ "C-T" = 35 °C), (T$_m$ ODN$^2$ "T-T" = 38 °C), and (T$_m$ ODN$^3$ "G-T" = 47 °C)[51], as well as other reported duplexes containing C-T, T-T, and G-T[52,54,55]. A weak interaction between C and T is further supported by the 10-fold faster Hg$^{II}$ binding of C-T versus T-T mismatches[51]. Taken together with the axial bending in our structure, these results are consistent with the fact that C-T mismatches are among the most thermodynamically destabilizing mismatches known in duplex DNA[52,54,55].

**Nucleobase-metal-nucleobase connectivity of C-Hg$^{II}$-T.** Adding three equivalents of Hg$^{II}$ (1.5 : 1.0 with respect to the number of mismatches) to duplex ODN$^1$ "C-T" caused disappearance of the mismatched imino resonance at 10.9 ppm (Fig. 3a and Supplementary Fig. 3)[51]. Similar results were obtained for the duplex ODN$^2$ "T-T" containing two T-T mismatches (Supplementary Fig. 3). These results reflect specific binding reactions, since the addition of Hg$^{II}$ to an analogous duplex containing two G-T wobble base pairs (ODN$^3$ "G-T") caused no such deprotonation (Supplementary Fig. 3). [$^1$H,$^1$H]-NOESY cross peaks between NH of thymidine and guanine residues, and between thymidine NH and H2 of adenine residues enabled assignment of all imino proton signals of ODN$^1$ "C-T" in both the presence and absence of Hg$^{II}$ (Supplementary Figs. 4, 5). The imino proton resonances of the thymidine residues flanking the C-T mismatch exhibited the largest changes in chemical shifts upon Hg$^{II}$ addition, giving a final spectrum similar to that of ODN$^2$ "T-T" containing the widely studied T-Hg$^{II}$-T base pairs (Supplementary Fig. 3, Supplementary Table 2)[41,42,46,48,49].

To characterize the structure(s) of C-Hg$^{II}$-T base pairs, we synthesized a $^{15}$N-labelled ODN$^{1*}$ "C*-T*" by synthetic incorporation of $^{15}$N-labelled C and T residues at positions 4 and 11 in an otherwise unlabelled duplex. The splitting of the $^1$H resonance at 10.9 ppm by $^{15}$N (88 Hz), and its disappearance upon adding Hg$^{II}$ (Fig. 3a) further confirmed its assignment as the mismatched thymidine NH-resonance. The five $^{15}$N-resonances in the absence of mercury were assigned by proton-coupled and proton-decoupled $^{15}$N NMR spectra, $^1J$ $^1$H,$^{15}$N coupling of thymidine N3-H and cytosine NH$_2$ by heteronuclear single quantum coherence (HSQC), and $^3J$ $^1$H,$^{15}$N- and $^2J$ $^1$H,$^{15}$N coupling

**Table 1 NMR restraints and statistics.$^a$**

|  | Apo duplex | Hg$^{II}$ duplex (major form) | Hg$^{II}$ duplex (minor form) |
|---|---|---|---|
| NOE-derived distance restraints$^b$ | 958 | 646 | 640 |
| Intra-nucleotide | 302 | 266 | 264 |
| Inter-nucleotide ($i$-$j$ = 1) | 566 | 306 | 302 |
| Long-range ($i$-$j$ = ≥ 2) | 90 | 74 | 74 |
| C-T, C-Hg$^{II}$-T | 250 | 158 | 148 |
| Repulsive | 0 | 0 | 0 |
| NOE restraints per residue | 34.21 | 23.07 | 22.86 |
| NOE violation >0.2 Å | 0 | 0 | 0 |
| Dihedral restraints$^{b,c}$ | 168 | 168 | 168 |
| Dihedral violations >5.0 ° | 0 | 0 | 0 |
| Hydrogen-bond restraints$^{b,c}$ | 60 | 62 | 62 |
| Planarity$^c$ | 24 | 24 | 24 |
| r.m.s.d (all heavy atoms vs. best structure) |  |  |  |
| Overall | 0.74 ± 0.26 Å | 1.21 ± 0.44 Å | 1.09 ± 0.36 Å |
| Helix | 0.83 ± 0.29 Å | 1.37 ± 0.48 Å | 1.21 ± 0.40 Å |
| C-T, C-Hg$^{II}$-T base pairs | 0.54 ± 0.21 Å | 0.76 ± 0.28 Å | 0.75 ± 0.29 Å |

$^a$Statistics are given for the 20 lowest energy structures from 200 calculated structures. $^b$Experimentally derived constraints (Supplementary Figs. 14, 16). $^c$Introduced constraints. The two additional hydrogen-bond restraints in the Hg$^{II}$-containing duplexes reflect N3-N3/N4 restraints for metal binding. The r.m.s.d. values are given as mean ± standard deviation. Source data are provided as a Source Data file

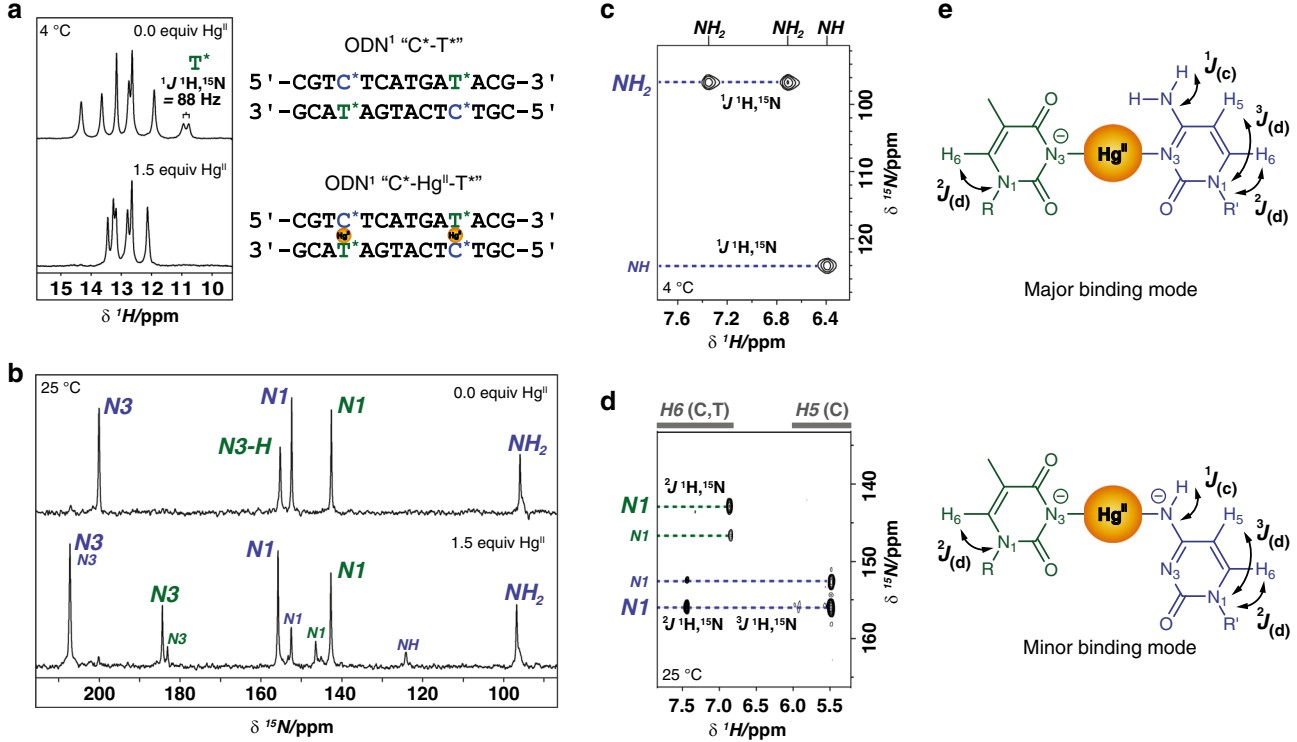

**Fig. 3** $^1$H and $^{15}$N NMR spectra of duplex DNA upon formation of C-Hg$^{II}$-T base pairs. **a** Imino proton region of $^1$H NMR spectra of ODN$^{1*}$ "C*-T*" in the absence and presence of 1.5 equiv of Hg$^{II}$ (where blue **C*** = $^{15}$N-labelled C, and green **T*** = $^{15}$N-labelled T). **b** $^{15}$N NMR spectra of ODN$^{1*}$ "C*-T*" before and after adding Hg$^{II}$. **c** $^1J$ $^1$H,$^{15}$N coupling of cytosine (N4)NH$_2$$^{major}$ and of cytosine (N4)NH$^{minor}$ according to [$^{15}$N,$^1$H]-HSQC. For full spectrum see Supplementary Fig. 9b. **d** Assignment of N1$^{major}$ and N1$^{minor}$ of thymidine and cytosine by $^3J$- and $^2J$-coupling between N1 and H5 and/or H6 in the long-range [$^{15}$N,$^1$H]-HSQC spectrum. For full spectra see Supplementary Fig. 10a. **e** Major- and minor coordination modes of Hg$^{II}$ bound to C-T mismatches. DNA samples contained 1.0 mM **b** or 0.5 mM **a**, **c**, and **d** of duplex DNA and 1.5 equiv of Hg$^{II}$ (relative to the mismatches present) in aqueous buffer (200 mM NaClO$_4$, 50 mM cacodylic acid in H$_2$O / D$_2$O (9:1) at pH = 7.8). Blue labels indicate $^{15}$N resonances of C* and green labels indicate $^{15}$N resonances of T*

between N1 and H5 and/or H6 by long-range [$^{15}$N,$^1$H]-HSQC (Supplementary Figs. 6, 7). After adding three equiv of Hg$^{II}$ (1.5 : 1.0 with respect to the number of mismatches), two sets of $^{15}$N-resonances were observed, corresponding to a "major" and a "minor" species in a 3:1 ratio (Fig. 3b). When adding only two equiv of Hg$^{II}$, these same signals were observed, in addition to those of unbound DNA, confirming that the minor species was not a result of any excess of Hg$^{II}$ (Supplementary Fig. 8). [$^{15}$N,$^1$H]-HSQC spectra allowed assignment of all $^{15}$N signals for both the major and minor complexes (Fig. 3c, d, Supplementary Figs. 9, 10). Disappearance of the N3-H cross peak in the $^1J$ [$^{15}$N,$^1$H]-HSQC spectrum confirmed deprotonation of thymidine N3 upon addition of 1.5 equiv of Hg$^{II}$ (Supplementary Figs. 7, 8). The downfield shift of thymidine N3$^{major}$ ($\Delta$ppm = + 29) and N3$^{minor}$ ($\Delta$ppm = + 28)[46,56] suggested that direct (N3)T-Hg$^{II}$ coordination was present in both binding modes (Fig. 3b, Supplementary Tables 3, 4). $^3J$ $^1$H,$^{15}$N- and $^2J$ $^1$H,$^{15}$N coupling of N1 to H5 and/or H6 observed in long-range [$^{15}$N,$^1$H]-HSQC spectra allowed for assignment of N1$^{major}$ and N1$^{minor}$ resonances, as well as H5 and H6 protons for cytosine and H6 of thymidine (Fig. 3d and Supplementary Fig. 10). The appearance of a single cross peak in the direct [$^{15}$N,$^1$H]-HSQC spectrum at 124/6.39 ppm indicated deprotonation of cytosine (N4)NH$^{minor}$ in the minor binding mode (Fig. 3c). The large downfield shift of cytosine (N4)NH$^{minor}$ ($\Delta$ppm = + 28, Fig. 3b, Supplementary Tables 3, 4) and a doublet observed in the [$^{15}$N,$^1$H]-proton-coupled HSQC ($J$ = 86 Hz, Supplementary Fig. 9c) further suggested displacement of one (N4)NH$_2$ proton

by Hg$^{II}$ and direct Hg$^{II}$ coordination to N4 in the minor species. An NOE-cross peak between (N4)NH$^{minor}$ to cytosine H5 (6.52/ 5.48 ppm at 25 °C) confirmed this $^{15}$N-resonance assignment (124 ppm) as being the deprotonated exocyclic amine of cytosine (Supplementary Fig. 11). $^3J$ $^1$H,$^{15}$N coupling of the $^{15}$N-signal at 207 ppm to (N4)NH$^{minor}$, as well as to (N4)NH$_2$$^{major}$ was observed in a band-selective, long-range [$^{15}$N,$^1$H]-HSQC (Supplementary Fig. 10b). This allowed assignment of the overlapping $^{15}$N-resonances of the N3$^{major}$ and N3$^{minor}$ of cytosine. It was still unclear, however, if Hg$^{II}$ was directly coordinated to cytosine N3$^{major}$.

To unambiguously determine metal-nucleobase connectivities[57], we monitored changes in the $^{15}$N NMR spectrum of ODN$^{1*}$ "C*-T*" upon addition of $^{199}$Hg-isotopically enriched (79 %) Hg(ClO$_4$)$_2$ (Fig. 4a). In the major binding mode, the N3-resonances for both cytosine and thymidine appeared as doublets, thereby revealing their direct coordination to Hg$^{II}$ (Fig. 4a). The large $^1J$ $^{15}$N,$^{199}$Hg coupling constant of 1095 Hz for thymidine Hg$^{II}$-N3$^{major}$ is consistent with T(N3)-Hg$^{II}$ binding reported for a T-Hg$^{II}$-T dinucleoside complex measured in $d6$-DMSO[57]. Cytosine N3$^{major}$-Hg$^{II}$ exhibited a much smaller coupling constant $^1J$ $^{15}$N,$^{199}$Hg = 114 Hz (Fig. 4a), consistent with a longer, weaker bond. A doublet with a coupling constant $^1J$ $^{15}$N,$^{199}$Hg = 1052–1063 Hz confirmed direct Hg$^{II}$ coordination to cytosine (N4)NH$^{minor}$ (Fig. 4a, b). This larger coupling constant is consistent with a stronger, shorter bond for Hg$^{II}$-C (N4) versus Hg$^{II}$-C(N3). Further support of our assignments was observed in $^1$H,$^{199}$Hg through-bond coupling with $^2J$ $^1$H,$^{199}$Hg

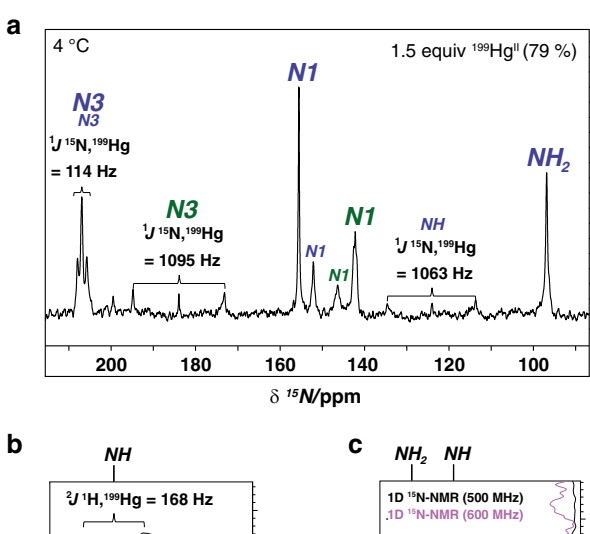

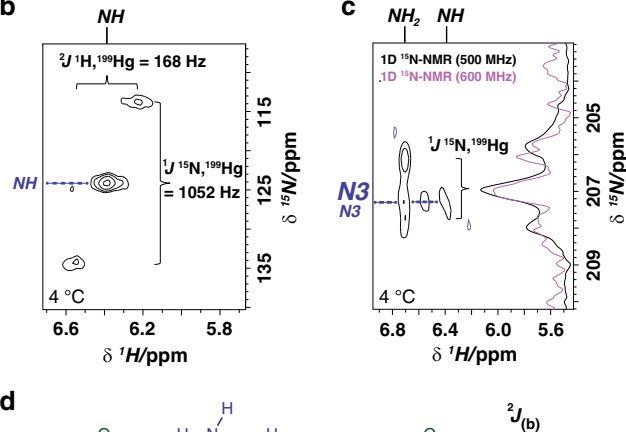

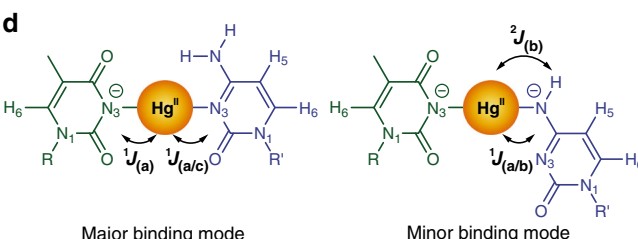

**Fig. 4** Nucleobase-metal-nucleobase connectivity of C-Hg$^{II}$-T base pairs. **a** $^{15}$N NMR of ODN$^{1*}$ "C*-T*" in the presence of 1.5 equiv of $^{199}$Hg-enriched Hg(ClO$_4$)$_2$ (79 % enriched see Supplementary Fig. 13). **b** $^1J$ $^{15}$N,$^{199}$Hg and $^2J$ $^1$H,$^{199}$Hg coupling according to [$^{15}$N,$^1$H]-HSQC. For full spectra see Supplementary Fig. 12b. **c** $^1J$ $^{15}$N,$^{199}$Hg coupling between cytosine N3$^{major}$ and $^{199}$Hg observed in the band-selective, long-range [$^{15}$N,$^1$H]-HSQC spectrum. For full spectrum see Supplementary Fig. 12c. **d** Summary of major- and minor coordination complexes in duplex DNA and the observed couplings. DNA samples contained 1 mM duplex DNA and 3 mM $^{199}$Hg-enriched Hg(ClO$_4$)$_2$. Blue labels indicate $^{15}$N resonances of C* and green labels indicate $^{15}$N resonances of T*

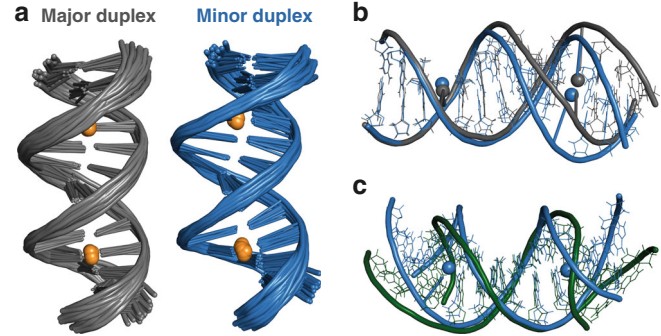

**Fig. 5** Solution structures of ODN$^1$ containing two C-Hg$^{II}$-T base pairs and comparisons with the metal-free duplex. **a** Side-view overlay of the 20 lowest energy conformations for major- (grey) and minor (blue) metallo duplex structures. The 20 lowest energy structures were aligned to one representative model of each duplex containing zero outliers in the geometric quality criteria: for the major duplex, #1 of PDB 6FY6, and for the minor duplex #6 of 6FY7. Hg$^{II}$ ions are depicted in gold. **b** Overlay comparison of these representative models of the major (grey) and minor (blue) duplexes. The duplexes were aligned by the end of each helix (C1, G2, T3, A26, C27, G28). The (N3)T-Hg$^{II}$-(N4)C connectivity in the minor duplex makes this metallo-base pair more solvent accessible than (N3)T-Hg$^{II}$-(N3)C, with a difference in solvent accessible surface area ($\Delta$SASA) = 41 Å$^2$ as calculated by PyMOL.[85] **c** Overlay comparison of representative minor metallo duplex (blue, model #6) as compared to metal-free duplex (green, model #1, PDB 6RLS). The duplexes were aligned by the center of each helix (A7, T8, A21, T22)

**Global structures of duplexes containing C-Hg$^{II}$-T.** Consistent with $^{15}$N NMR spectra (Figs. 3, 4), $^1$H NMR experiments confirmed the presence of two $C_2$ symmetrical species (Supplementary Fig. 14). Proton resonance assignments and modelling for each duplex were conducted exactly the same as for the metal-free duplex. Aside from the first and last residues of the duplexes (C1 and G14), all signals of the major and minor metallo duplexes were well resolved and the sequential walk proceeded through the entire sequence (Supplementary Figs. 14, 15). Models for major and minor duplex were constructed based on 646 and 640 NOE-derived conformationally restrictive distance restraints, respectively (Table 1). Superimposition of the 20 lowest energy structures of 200 computed structures gave an overall root mean square deviation (r.m.s.d) of all heavy atoms of 1.21 ± 0.44 Å for the major duplex and 1.09 ± 0.36 Å for the minor duplex (Fig. 5a, Table 1).

As compared to the major duplex, the minor duplex exhibits more axial bending and a deeper, narrower major groove (Fig. 5b). The differences between the metal-bound and metal-free structures are also the greatest for the minor duplex (Fig. 5c). Inspection of the 20 lowest energy models for all three structures (apo, major metallo, and minor metallo) revealed a high frequency (65–100 %) of an unusual, O4′-endo sugar pucker at the cytosine residue of the C-T and C-Hg$^{II}$-T base pairs (Supplementary Table 6). Direct support for this was observed in the $^3J$ H1′,H2′ coupling constant for the Hg$^{II}$-coordinated cytosine residue ($^3J$ H1′,H2′ = 6.5 Hz) in the major structure (Supplementary Table 5, Supplementary Fig. 17). Using the Karplus equation (Supplementary Equation 1)[61] a dihedral bond-angle of $\Phi_{1'2'}$ = 142° was calculated, which is in excellent agreement with dihedral angles observed in the solution structure models ($\Phi_{1'2'}$ = 131°) (Supplementary Fig. 18, Supplementary Equation 1)[61,62]. However, the apparent O4′-endo sugar puckers present in all three structures likely reflect the averaged conformations of rapidly interconverting C2′- and C3′-endo sugar puckers that occur much faster than the time scale of these NMR measurements[63–67].

= 168 Hz in the [$^{15}$N,$^1$H]-HSQC (Fig. 4b), a value that was similar to reported $^1$H,$^{199}$Hg coupling across nitrogen in unrelated systems measured in $d6$-DMSO or CDCl$_3$[58–60]. Splitting of the correlation between cytosine N3$^{major}$ and (N4) NH$_2$$^{major}$ in the band-selective, long-range [$^{15}$N,$^1$H]-HSQC further confirmed our assignment of N3$^{major}$ of cytosine (Fig. 4c). These data provide $^1J$ $^{15}$N,$^{199}$Hg coupling constants in aqueous solutions, as well as cytosine N3,Hg$^{II}$ and cytosine N4,Hg$^{II}$ $^1J$ $^{15}$N,$^{199}$Hg coupling constants. Taken together, these results revealed that Hg$^{II}$ binds to cytosine via two distinct coordination modes in solution. The major species contained two identical and unambiguous (N3)T-Hg$^{II}$-(N3)C base pairs, and the minor species two identical and unambiguous (N3)T-Hg$^{II}$-(N4)C base pairs. No evidence for a lower symmetry duplex containing one of each type of base pair was observed in any NMR experiment.

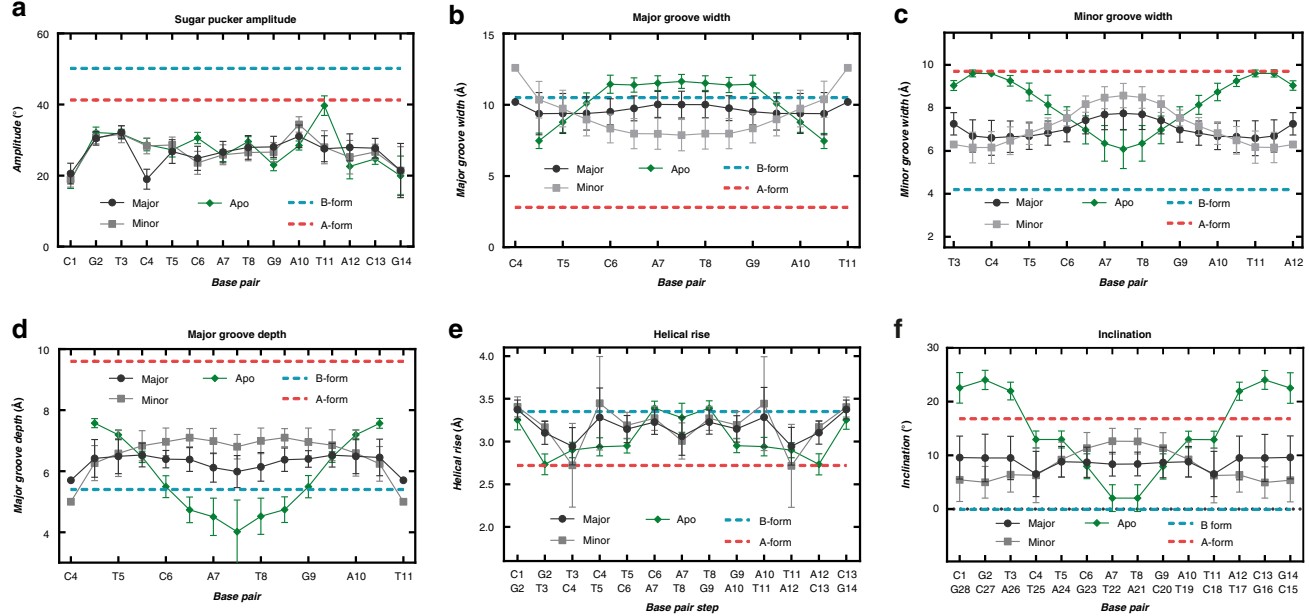

**Fig. 6** Selected base pair and base pair step parameters of apo-, Hg[II]-bound major-, and Hg[II]-bound minor duplex structures. Parameters of apo (green diamonds, PDB 6RLS), Hg[II]-bound major (black circles, PDB 6FY6) and Hg[II]-bound minor (grey squares, PDB 6FY7) duplex structures were calculated using Curves+.[86] **a** Sugar pucker amplitude. **b** Major groove width. **c** Minor groove width. **d** Major groove depth. **e** Helical rise. **f** Inclination. The reference values for standard A- (red) and B-form (blue) duplex DNA were taken from (ref 86), which analyzed crystal structures of A-DNA (PDB 1d13[88]) and B-DNA (PDB 1bna[66]) dodecamers. Reported values represent the mean and standard deviation of the 20 lowest energy conformations for each structure. For additional analyses of base pair parameters see Supplementary Figs. 19 –22. Source data are provided as a Source Data file

Support for this conclusion can be found in the analyses of global parameters, where the sugar pucker amplitudes throughout all three duplexes were much smaller than both A-form and B-form duplexes (Fig. 6a). In contrast, nearly all other structural parameters including groove dimensions, twist and rise gave values intermediate between A- and B-form duplexes (Fig. 6b–f, and Supplementary Figs. 19–22)[68].

**Dynamics of major—minor metallo duplex interconversion**. The presence of two individual sets of NMR signals, together with exchange cross peaks and exchange-mediated cross peaks between them (Supplementary Figs. 23–26 and Supplementary Figs. 36–38) indicated the presence of conformational changes with rates suitable for determination by standard NMR methods. To investigate the dynamic changes in local metal ion coordination, z-z exchange [15N,1H]-HSQC spectra were measured[69]. With increasing delay time ($t_m$), new exchange signals appeared (Fig. 7a, Supplementary Fig. 23). Global fitting of integrated peak volumes versus exchange delay times (Supplementary Equations 6–9) furnished rate constants of $k_1 = 3.5\,\mathrm{s^{-1}}$ and $k_{-1} = 7.7\,\mathrm{s^{-1}}$ for the forward and reverse reactions of nucleobase-metal-nucleobase isomerization, respectively (Fig. 7a).

Single exchange signals in ROESY spectra were observed for the Hg[II]-bound nucleosides as well as for sugar- and aromatic proton signals of various residues throughout the duplex (Supplementary Fig. 24). The same exchange signals were found in the [1H,1H]-TOCSY spectrum in the H1′/H5 region (~5.0–6.2 ppm, Supplementary Fig. 25). Within that region, no protons belonged to the same spin system. The observed cross peaks therefore occurred by exchange rather than through-bond coupling. Exchange-mediated cross peaks were observed in the aromatic → H2′/H2″ region of the [1H,1H]-NOESY spectrum (Supplementary Fig. 26), confirming the global nature of the conformational change. To evaluate the potential impact of variable ionic strength on the exchange rates, samples of the

metallo duplex were prepared in the presence of 50 mM, 200 mM and 500 mM NaClO4. The overall exchange rates of interconversion between the two structures decreased with increasing ionic strength, but the ratio of the two structures remained the same (Supplementary Figs. 27–31). Likewise, pH-dependent measurements revealed slower overall exchange rates with increasing pH from pH 6–9 while maintaining the same ratio (Supplementary Figs. 32–35). The lower rates of exchange correlate with the increasing persistence length and therefore rigidity of the duplex with increasing pH and ionic strength[70,71].

To determine rate constants of global interconversion of the two duplexes, we measured [1H,1H]-NOESY spectra with various mixing times ($t_m$) using samples prepared in the presence of 200 mM NaClO4 at pH = 7.8. Selected exchange cross peaks ('Aa' and 'aA', Supplementary Fig. 37) and exchange-mediated NOE cross peaks ('Ab' and 'aB', Supplementary Fig. 38) at residues throughout the duplex were integrated, normalized to signal intensity at mixing time = 200 ms, and plotted as a function of mixing time ($t_m$) (Supplementary Table 7, Supplementary Figs. 39–42). For definitions of peak labels, see Supplementary Fig. 36. Exchange cross peaks were fit to Supplementary Equation 12 to determine the sum of rate constants $k_1 + k_{-1}$ (Fig. 7b, Supplementary Fig. 39 and Supplementary Table 7). For signals having sufficient resolution of the diagonal peaks ('AA' and 'aa') for integration purposes, exchange cross peaks were normalized by aA ($t_m$) / (AA ($t_m$) + aA ($t_m$)) and fit to Supplementary Equation 14 to determine individual rate constants $k_1$ and $k_{-1}$ (Fig. 7b and Supplementary Fig. 41). Dividing exchange-mediated cross peaks ('Ab' and 'aB') by the sum with their corresponding NOE cross peak ('AB' and 'ab') allowed the determination of rate constants $k_1$ and $k_{-1}$ by fitting to Supplementary Equations 18 and 19 (Supplementary Fig. 42). The sum of rate constants $k_1 + k_{-1}$ determined for the global conformational change according to exchange cross peaks (10.5–15.0 s⁻¹), as well as the individual rate constants $k_1$ (3.8–5.2 s⁻¹) and $k_{-1}$ (7.5–9.7 s⁻¹) independently determined

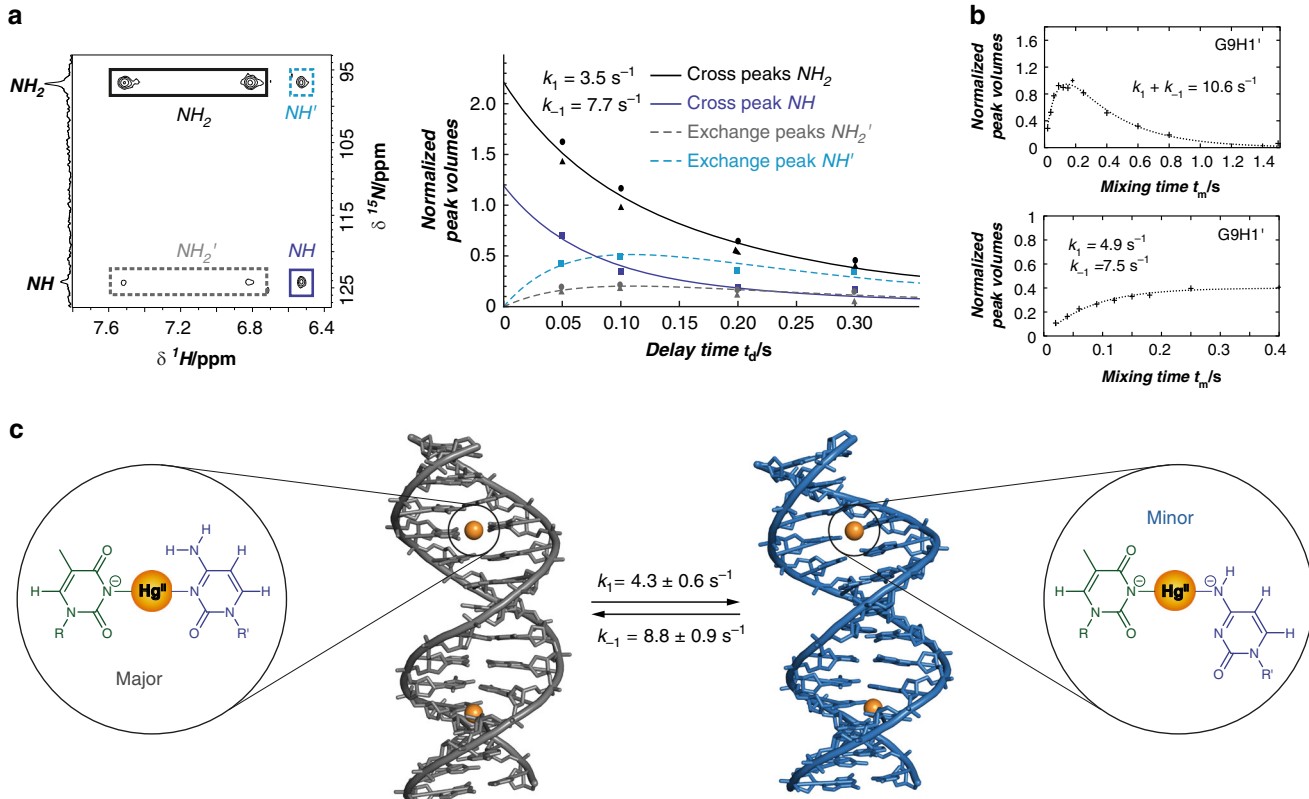

**Fig. 7** Dynamic isomerization of duplex DNA containing two C-Hg$^{II}$-T base pairs. **a** Dynamic changes in local metal ion coordination. Rate constants of interconversion were determined by plotting changes in cross peaks ($NH_2$ and $NH$) and exchange cross peaks ($NH_2'$ and $NH'$) observed in z-z exchange [$^{15}$N,$^1$H]-HSQC spectrum as a function of different delay times. Rate constants of interconversion were determined by global fitting of integrated peak volumes versus exchange delay times using Supplementary Equations 6-9, where $R_1$, $k_1$, and $k_{-1}$ were constrained as being equal for all four curves with the assumption that all proton- and nitrogen atoms have the same relaxation rate ($R_1$) and that the $NH$ proton is converted equally to both $NH_2$ protons during the exchange. **b** Representative example (G9H1') of duplex interconversion rate constants as determined by monitoring changes in exchange cross peaks and exchange-mediated NOE cross peaks in [$^1$H,$^1$H]-NOESY spectra with various mixing times ($t_m$). At longer mixing times, a decrease in signal intensity was observed due to auto relaxation ($R_1$). The sum of interconversion rate constants ($k_1 + k_{-1}$) was determined by fitting changes in cross peak intensity vs mixing time ($t_m$) (Supplementary Equation 12). The individual rate constants of interconversion ($k_1$ and $k_{-1}$) were determined by fitting (Supplementary Equation 14) cross peak intensities normalized by the intensities of diagonal peaks versus mixing times. See Supplementary Figs. 37, 38 for [$^1$H,$^1$H]-NOESY spectra, and Supplementary Figs. 39–42 for the same types of analyses at other positions. **c** Summary of major- and minor duplex structures, structures of the metallo base pairs they contain, and the rate constants of their exchange. Rate constants values are given as mean ± standard deviation from 5 exchange cross peaks and exchange-mediated NOE cross peaks. The Hg$^{II}$-ligand complexes present in the major species are net positive (+1), whereas those of the minor species are neutral. The combined contributions of more favourable electrostatic interactions and near-canonical B-form helical structure present in the major species likely outweigh the stronger ligand–metal–ligand binding interactions present in the minor duplex

from exchange cross peaks and exchange-mediated cross peaks, were all in excellent agreement with rate constants determined for changes in metallo base pair structure ($k_1 = 3.5$ s$^{-1}$, $k_{-1} = 7.7$ s$^{-1}$, Fig. 7a, Supplementary Table 7). Together with the absence of a third, lower symmetry duplex containing one of each type of metallo-base pair, these results revealed that dynamic changes in local metal-ligand isomerization were coupled to the global interconversion of the two duplex structures (Fig. 7c).

**Hg$^{II}$-induced, conformational switching from B- to A-form.** The long-range coupling (>20 Å) between the metal centres as well as the increased A-form characteristics in the centre region of the duplex upon metal binding, suggested that placing numerous C-T mismatches throughout a repetitive duplex sequence could facilitate a global B → A helical transition upon adding Hg$^{II}$. To test this possibility, we introduced C-T mismatches into $(G_4C_4)_n$ type DNA sequences that are known to exhibit partial A-form characteristics[50,72–74]. To supress formation of intramolecular G-quadruplex structures that would

otherwise interfere with intermolecular duplex formation of such repetitive sequences, we designed a small library of 120-mer DNA hairpins (Supplementary Table 8) containing the tetraloop sequence cGCTAg that is known to stabilize both RNA and DNA hairpins[75]. To fold the hairpins, dilute solutions of DNA (1 µM) were heated (95 °C, 5 min) and rapidly cooled on ice at 0 °C. Samples were then incubated with 0.0 or 1.5 equiv of Hg$^{II}$ (relative to number of C-T mismatches present) at 25 °C for 3 h prior to their analysis. Gel electrophoresis revealed clean, intramolecular hairpin formation for most sequences in both the presence and absence of Hg$^{II}$, including our hit ODN[13] (Supplementary Fig. 43). To screen for the induction of A-form structures by Hg$^{II}$, we used a fluorescent aminoglycoside binding assay[74]. Aminoglycoside antibiotics exhibit a general selectivity for binding A-form over B-form helices[6]. Changes in the fluorescence anisotropy of a 40 nM solution of a Neomycin-BODIPY conjugate "Neo-BODIPY"[76] were therefore measured in the presence and absence of each DNA (600 nM) pre-treated with Hg$^{II}$ (0.0 or 1.5 equiv per C-T mismatch). No changes in

anisotropy were observed upon addition of all hairpins in the absence of Hg$^{II}$, however, the pre-incubation of ODN$^{13}$ with Hg$^{II}$ caused a 3.5-fold increase in fluorescence anisotropy of Neo-BODIPY (Table 2). Titration of the ODN$^{13}$-Hg$^{II}$ complex into solutions of Neo-BODIPY revealed an apparent dissociation constant ($K_d$) = 1.4 ± 0.7 μM (Fig. 8a). This value is similar to the values reported for binding of neomycin to A-form, duplex RNA[77]. The ternary complex formed between Neo-BODIPY and ODN$^{13}$-Hg$^{II}$ was disrupted by the addition of unlabelled neo-mycin B (Supplementary Fig. 44), as well as *N*-acetylcysteine that sequesters Hg$^{II}$ (Supplementary Fig. 45). These results demon-strate the reversibility of Neo-BODIPY binding, a lack of sig-nificant impact by the BODIPY tag, as well as the switch-like (on/ off) effect of Hg$^{II}$ binding to ODN$^{13}$. The analogous hairpin containing T-A base pairs (ODN$^{14}$) in place of T-C mismatches exhibited no such behaviour (Table 2). These observations are confirmed using CD spectroscopy.

The (G$_3$C$_3$)$_n$-containing hairpins ODN$^4$ and ODN$^5$ exhibited CD spectra consistent with previous publications[72], having a double maximum at 260 nm and 280 nm (Fig. 8b and Supplementary Fig. 46) that are thought to reflect a mixture of

A-form and B-form-like stacking of the guanine and cytosine nucleobases, respectively. The addition of Hg$^{II}$ had little-to-no impact on the CD spectra of hairpins ODN$^{4–12}$ or ODN$^{14}$ (Fig. 8b and Supplementary Fig. 46). However, addition of Hg$^{II}$ to hairpin ODN$^{13}$ caused changes in its CD spectrum indicative of a global B- to A-form helical transition (Fig. 8c and Supplementary Fig. 47a)[78]. This Hg$^{II}$-induced conformational change exhibited a 1:1 stoichiometry between Hg$^{II}$ and the number of C-T mismatches present (Supplementary Fig. 47b), was extremely rapid (<30 s to complete) and fully reversible upon addition of *N*-acetylcysteine (Supplementary Fig. 47c). Given the relatively low G-quadruplex propensity of sequence ODN$^{13}$, we were successful in preparing the corresponding intermolecular duplex lacking a hairpin turn. This simple duplex "ODN$^{13}$ ds" also exhibited a reversible, Hg$^{II}$-inducted switching between global B- and A-form helices (Supplementary Fig. 48). By alternating between the addition of Hg$^{II}$ and *N*-acetylcysteine, the helical switching cycle from B- to A-form, and A-form to B-form could be repeated more than 10 times on the same DNA (Supplementary Fig. 49).

## Discussion

Here we report solution structures of C-Hg$^{II}$-T base pairs. In addition to providing fundamental insights into C-Hg$^{II}$-T bind-ing and dynamics, these results provide $^1J$ $^{15}$N,$^{199}$Hg and $^2J$ $^1$H,$^{199}$Hg coupling constants. Prior to these studies, little or no such information was available for these coupling constants in water[46,47,58,79]. Large differences in $^1J$ $^{15}$N,$^{199}$Hg coupling con-stants for complexes containing (N3)C-Hg$^{II}$ (114 Hz) versus (N4) C-Hg$^{II}$ (1052 Hz) will provide a basis for future computational studies that address the relationships between coupling constants, ligand ionization and metal binding[57].

Aside from the gain/loss of a proton from (N4)C, the major/ minor duplexes of ODN$^1$ bound to Hg$^{II}$ are constitutional iso-mers with respect to each other. Isomerization-coupled con-formational exchange between these structures was relatively slow on the chemical-shift NMR time scale, yet it was fast enough to allow direct characterization by monitoring changes in exchange cross peaks as a function of mixing time ($t_m$). The rate constants of nucleobase-metal-nucleobase isomerisation ($k_1 = 3.5$ s$^{-1}$ and $k_{-1} = 7.7$ s$^{-1}$) were nearly identical to rate constants indepen-dently measured for the global conformational exchange of the two duplex structures ($k_1 = 4.3 ± 0.6$ s$^{-1}$, $k_{-1} = 8.8 ± 0.9$ s$^{-1}$). No evidence for a third duplex containing one of each type of metal-base pair was observed, giving further support for coupling between the metal centres via global conformational exchange.

**Table 2 DNA hairpin repeat sequences[a] and fluorescence anisotropy of Neo-BODIPY in the presence of each DNA with and without Hg$^{II}$.[b]**

| Name | DNA repeat sequence | no Hg$^{II}$ | + Hg$^{II}$ |
|------|---------------------|--------------|-------------|
| – | (Neo-BODIPY only) | 0.05 ± 0.01 | 0.05 ± 0.01 |
| ODN$^4$ | [**C**GGGC**C**CGGGCC]$_{4.5}$ | 0.05 ± 0.01 | 0.06 ± 0.01 |
| ODN$^5$ | [**T**GGGCCC**C**GGGGCCCC]$_{3.5}$ | 0.05 ± 0.01 | 0.06 ± 0.01 |
| ODN$^6$ | [**C**GGG]$_{14}$ | 0.05 ± 0.02 | 0.05 ± 0.01 |
| ODN$^7$ | [**T**C**T**GGGC]$_8$ | 0.04 ± 0.01 | 0.05 ± 0.02 |
| ODN$^8$ | [**T**GG**T**GCC**C**CCGG]$_5$ | 0.05 ± 0.01 | 0.04 ± 0.01 |
| ODN$^9$ | [**T**GGG**T**GG**T**CGC]$_5$ | 0.05 ± 0.01 | 0.05 ± 0.01 |
| ODN$^{10}$ | [**C**GGG**T**GG**C**CGC]$_5$ | 0.04 ± 0.01 | 0.05 ± 0.01 |
| ODN$^{11}$ | [**T**GGG**C**GG**T**CGC]$_5$ | 0.05 ± 0.01 | 0.05 ± 0.01 |
| ODN$^{12}$ | [**T**CC**T**CGG**T**GGC]$_5$ | 0.04 ± 0.02 | 0.05 ± 0.01 |
| ODN$^{13}$ | [**T**GG**T**CCC**T**CGG]$_5$ | 0.04 ± 0.01 | 0.14 ± 0.01 |
| ODN$^{14}$ | [*TGGTCCCTCGG*]$_5$ | 0.04 ± 0.01 | 0.05 ± 0.01 |

[a]Bold bases indicate **C-T** mismatches. Italic bases in ODN$^{14}$ indicate T-A base pairs. [b]All samples contained 40 nM of Neo-BODIPY, 600 nM of DNA, and 0 or 1.5 equiv of Hg$^{II}$ per C-T mismatch in an aqueous buffer containing 200 mM NaClO$_4$ and 50 mM cacodylic acid (pH = 7.8). Averaged anisotropy values and standard deviations of three independent measurements are shown. Source data are provided as a Source Data file

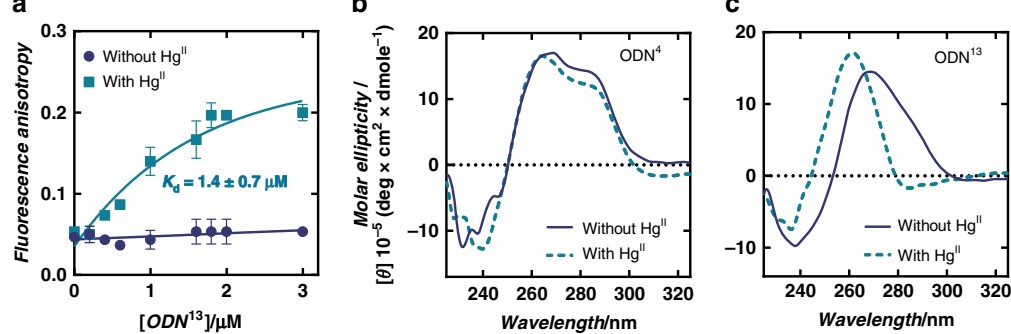

**Fig. 8** Fluorescence anisotropy and CD spectra reveal global B-form to A-form helical transition upon addition of Hg$^{II}$. **a** Changes in fluorescence anisotropy ($\lambda_{ex}$ = 480 nm, $\lambda_{em}$ = 515 nm) of Neo-BODIPY (40 nM) upon addition of hairpin ODN$^{13}$ prepared in the presence (light blue) and absence (dark blue) of Hg$^{II}$. The values represent mean and standard deviation of three independent measurements. C-Hg$^{II}$-T base pairs were formed by pre-incubation of the DNA with 1.5 equiv Hg$^{II}$ for 3 h prior to addition of Neo-BODIPY. **b** CD spectra of hairpin ODN$^4$ (1 μM), or, **c** CD spectra of hairpin ODN$^{13}$ (1 μM) in the presence and absence of 1.5 equiv of Hg$^{II}$. All samples were prepared in an aqueous buffer containing 200 mM NaClO$_4$ and 50 mM cacodylic acid (pH = 7.8). The number of Hg$^{II}$ equiv are relative to the number of mismatches present. Source data are provided as a Source Data file

The forward and reverse rate constants of this process were ~$10^3$-fold faster than dissociation of $Hg^{II}$ from C-$Hg^{II}$-T base pairs[51]. The $Hg^{II}$ ions therefore remained bound to the DNA during multiple structural exchanges. Given the high quality of the structures and dynamics reported here, this system provides a highly attractive model for the development of molecular dynamics simulations aimed at elucidating the pathways of dynamic conformation exchange processes[80].

Consistent with our structure of the minor duplex (Figs. 4, 5), crystals of a short, A-form DNA sequence also contained two C-$Hg^{II}$-T base pairs having (N3)T-$Hg^{II}$-(N4)C connectivity[50]. However, the global A-form structure observed in the crystal structure was inconsistent with our circular dichroism (CD) data collected in solution using five different, slightly longer (14 to 21-mer) duplexes containing one or two C-$Hg^{II}$-T base pairs[51]. Together with the observation that (N3)T-$Hg^{II}$-(N3)C binding present in our major structure is associated with more B-form character, and the (N3)T-$Hg^{II}$-(N4)C present in the minor structure is associated with more A-form character, a general picture is suggested. (N3)T-$Hg^{II}$-(N3)C binding and B-form structure is likely dominant in most DNA sequences, in direct analogy with high resolution structures of T-$Hg^{II}$-T base pairs[49]. However, the 10-fold larger $^1J$ coupling constants ($^{15}N,^{199}Hg$) observed for (N4)C-$Hg^{II}$ (1052 Hz) connectivity in the minor duplex as compared to the (N3)C-$Hg^{II}$ (114 Hz) in the major structure indicate that stronger nucleobase-metal-nucleobase binding interactions are present in the minor structure. The minor duplex, however, also exhibits a greater degree of structural perturbation away from a canonical B-form duplex that likely destabilizes the complex as compared to the major form. The lower bonding energy of (N3)T-$Hg^{II}$-(N3)C versus (N3)T-$Hg^{II}$-(N4)C is therefore compensated by the overall higher stability of B-form versus A-form in the major and minor structures, respectively. However, in situations that favour A-form (crystallization, dehydration, G/C-rich DNA sequences, etc.) the stronger (N3)T-$Hg^{II}$-(N4)C coordination mode can dominate. The exact causal relationship between global helical conformation and metallo-base pair connectivity is currently unclear. An X-ray study involving a short A-form DNA containing two (N3)T-$Hg^{II}$-(N4)C base pairs suggested the combined effects of base pair geometry, neighbouring base effects, and a bridging water molecule in the minor groove were invovled[50].

For both of our metal-bound structures in solution, the amount of A-form character was greatest in the centre of each helix, suggesting a medium-range influence of C-$Hg^{II}$-T formation. In contrast, the opposite pattern was observed in the structure of the metal-free duplex, having the greatest B-form characteristics in the centre of the helix. Together with the long-range coupling (>20 Å) between the two metallo-base pairs via a global conformational exchange, these results suggested that placing numerous C-T mismatches throughout a repetitive duplex sequence could support a global B-form → A-from helical transition upon addition of stoichiometric $Hg^{II}$. Indeed, we were able to identify one such sequence exhibiting a fully-reversible switching cycle from B- to A-form, and A-form to B-form by tandem additions of $Hg^{II}$ and N-acetylcysteine (Supplementary Fig. 49). Both transitions were complete in <30 s, and could be repeated more than 10 times. While numerous examples of local A-form perturbations caused by DNA-protein and DNA-small molecule binding interactions have previously been reported[3–9], the previous examples of global B-form → A-from helical transitions involved global dehydration of the duplex[1,2,78]. Here the global B- to A-form helical transition was a result of discrete, reversible metal binding. In addition to its broad implications in structural biology and biochemistry, this type of A/B-form helical

switching can be potentially be utilized in the development of advanced DNA-based materials and devices[24–31].

## Methods

**Sample preparation.** For Fig. 2, duplex DNA (0.3 mM) was prepared by dissolving 0.6 mM of the self-complementary sequence in an aqueous solution of $NaClO_4$ (50 mM, 90:10 $H_2O/D_2O$) and the pH was adjusted to pH = 7.8 by addition of an aqueous solution of NaOH. The sample was annealed by heating to 95 °C for 5 min and slow cooling to room temperature over 4 h. For Figs. 3, 4, 7, duplex DNA (0.5–1.0 mM) was prepared by dissolving 1.0 mM and 2.0 mM of the self-complementary sequence in aqueous buffer (200 mM $NaClO_4$, 50 mM cacodylic acid in $H_2O$ / $D_2O$ (9:1) at pH = 7.0), heating to 95 °C for 5 min, and slowly cooling to room temperature over 4 h. $Hg(ClO_4)_2$ was added, and the pH adjusted to pH ≈ 7.8 by addition of an aqueous solution of NaOH. For Figs. 5, 6, duplex DNA (0.4 mM) was prepared by dissolving 0.8 mM of the self-complementary sequence in an aqueous solution of $NaClO_4$ (50 mM, 90:10 $H_2O/D_2O$) and the pH was adjusted to pH = 7.8 by addition of an aqueous solution of NaOH. The samples were annealed as described above, mixed with $Hg^{II}$ (1.5 equiv with respect to mismatch), the pH re-adjusted to pH ≈ 7.8, and the sample was treated with Chelex-100 (BIO-RAD) for 10 min to remove excess $Hg^{II}$[143]. Samples measured in $D_2O$ were prepared the same way and then lyophilized, dissolved in 99.9% $D_2O$ and the pD was adjusted to ~7.4 by addition of a solution of NaOD in 99.9% $D_2O$. Samples measured at 4 °C were equilibrated at 4 °C for 15 min prior to measuring.

**NMR spectra measurements.** $^1H$ NMR spectra were recorded on a Bruker Avance II 500 MHz spectrometer equipped with a TXI z-axis gradient probe head using excitation sculpting for water suppression. Proton chemical shifts were referenced to the water line at 4.70 ppm. The spectra were processed with a line broadening factor of 10 Hz. 1D $^{15}N$-NMR spectra were recorded on a Bruker Avance II 500 MHz spectrometer equipped with a BBO z-axis gradient CryoProbe at 4 °C or 25 °C using either inverse gated or no proton decoupling. The spectra were processed with a line-broadening factor of 10 Hz. [$^{15}N,^1H$]-HSQC spectra were recorded at 4 °C or 25 °C on a Bruker Avance II 500 MHz spectrometer equipped with a BBO z-axis gradient CryoProbe, Bruker Avance 600 MHz spectrometer equipped with TCI z-axis gradient CryoProbe or on a Bruker Avance 700 MHz spectrometer equipped with TXI z-axis gradient CryoProbe. The INEPT times were set to select for a 90 Hz coupling for $^1J$ [$^{15}N,^1H$]-HSQC's, 20 Hz coupling for long-range [$^{15}N,^1H$]-HSQC spectra, and to 25 Hz for band-selective long-range [$^{15}N,^1H$]-HSQCs. Water flip-back pulses together with the WATERGATE method were used for water suppression. Proton chemical shifts were referenced to the water line at 4.70 ppm at 4 °C and $^{15}N$ chemical shifts were indirectly referenced against $^1H$ using Ξ = 0.101329118[81]. For solution structure determination, non-exchangeable resonances were assigned from [$^1H,^1H$]-NOESY spectra measured in $D_2O$ (4 °C and 25 °C and mixing times of 60 ms and 250 ms). Exchangeable protons were assigned from [$^1H,^1H$]-NOESY spectra measured in $H_2O/D_2O$ (90:10) (4 °C, 150 ms mixing time). Spectra were recorded on a Bruker Avance 600 MHz spectrometer equipped with a TCI z-axis gradient CryoProbe or on a Bruker Avance 700 MHz spectrometer equipped with TXI z-axis gradient CryoProbe. For additional information see supporting information.

**NMR solution structure calculations.** The integrated peak volumes from a representative [$^1H,^1H$]-NOESY spectrum (mixing time = 250 ms) measured at 25 °C were calibrated to distances using CALIBA macro in DYANA[82]. The NOE signals were grouped into four categories: (i) strong (1.8–3.0 Å), (ii) medium (1.8–4.5 Å), (iii) weak (3.0–6.0 Å), and very weak (4.0–7.0 Å). Structure calculations were performed with XPLOR-NIH 2.46 using standard implemented force field parameters[83]. For introduced restraints for the calculations see supporting information. Starting from a strand generated based on the sequence of nucleoside residues, 2000 structures were calculated based on NOE-, dihedral-, planarity-, and H-bond distance restraints using simulated annealing. The 20 lowest energy structures were selected and used for further refinement using additional RAMA and ORIE database terms. 200 refined structures were calculated and the 20 lowest energy structures were visualized and analyzed. Root mean square deviation (r.m.s. d.) were calculated using MOLMOL[84] and duplexes were visualized using PyMOL[85]. Base-pair parameters were determined using Curves +[86] or 3DNA[87].

## Data availability

Structures of the metal free (PDB 6RLS), $Hg^{II}$-bound major (PDB 6FY6) and $Hg^{II}$-bound minor (PDB 6FY7) duplex structures have been deposited in the Protein Data Bank (https://www.rcsb.org/). All other data generated and analyzed in this study are included in this article, supplementary information, source data file, and are also available from the authors upon reasonable request. Source data for Table 1, Table 2, Figs. 6, 7, 8, and Supplementary Figs. 15, 19–22, 28, 29, 31, 33–35, 39–44, 45, 47, 49 are provided in the Source Data file.

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

## Acknowledgements
We thank Prof. Oliver Zerbe for critical reading of this manuscript, Dr. Thomas Fox for NMR support, and Dr. Laurent Bigler and Urs Stalder for MS support. We thank Prof. Yitzhak Tor for his generous gift of Neo-BODIPY. Financial support was provided by the University of Zurich and the Swiss National Science Foundation (grant #165949 to N.W.L.).

## Author contributions
O.S. and N.L. conceived the project. O.S.; S.Ju.; S.Jo.; A.K.; R.S.; and N.L. contributed to experimental design and manuscript editing. O.S.; S.Ju.; S.Jo.; and A.K. collected and analysed the data. O.S. and N.L. wrote the manuscript.

## Competing interests
The authors declare no competing interests.
