## [Peer Review File · Nature Communications]

Reviewers' comments:

Reviewer #1 (Remarks to the Author):

Luedtke and co-workers reported in the manuscript Dynamic isomerization of a metallated double helix preparation and NMR measurements of novel metallo-DNA duplex including two C-Hg(II)-T base pairs separated by six normal Watson-Crick base pairs. This work represents outstanding contribution to the research of metallo-nucleic acids within the field of general metallo-organic chemistry.

The modification of nucleic acids has been recently recognized important namely in connection with development of functional molecules and advanced nano-materials. In that regard, the metallo-DNAs including "all-natural" nucleotides are particularly attractive because the platform enables relatively straightforward production of DNA molecules that can be functionalized upon adding of suitable metal. While the metal binding to several base pairs was recently determined only very little is known about conformational dynamics of metallo-DNAs. That is why I consider the work by Luedtke and co-workers significant for the field; new metal-binding within C-Hg(II)-T pair was characterized unambiguously and in detail, local conformational dynamics of the C-Hg(II)-T involving "major" and "minor" metal binding was determined inclusive respective rate constants, and particularly, the conformational dynamics of the tandem metallo-base pairs within metallo-DNA was determined.

Astonishingly, the conformational dynamics of the two C-Hg(II)-T base pairs was concerted, so that only "major" or "minor" tandem occurs at once. Plausible validation of the long-distance coupling of the dynamical states within C-Hg(II)-T pairs mediated by nucleic acid backbone is unique even in the context of general structural biochemistry. Moreover, the concerted dynamics of metallo-base pairs induced concerted exchange of B and A classes of the backbone conformations. I do consider the discovery of coupled local dynamics within tandem C-Hg(II)-T with the global dynamics of whole metallo-DNA probably the most significant outcome of this work. It can have impact on progress within the field including future applications. Similar complex studies on metallo-nucleic acids are very scarce and I am not aware of any other that provides such a complex picture. This work demonstrated that dynamics of the 3D architecture of metallo-DNA is well-controllable by sequence programming of metallo-base pairs. The metallo-DNA thus can be used as template to study number of biochemical processes, to benchmark other spectroscopic measurements and to develop/calibrate force-fields for molecular dynamics simulations. May the authors consider to highlight this fact already in the title? (e.g.: Concerted dynamics of metallo-base pairs and whole metallo-DNA due to the metal isomerization (?))

I would like to address one issue concerning sustainability of the metal-binding/conformational dynamics due to pH.

The NMR measurement was carried out at pH 7.8. When considering the Major and Minor metal bindings, the effect of pH may affect their dynamical exchange.

Can the authors explain whether the effect on conformational dynamics due to different pH can be assumed?

If yes, the pH-switch or pH-tuner of conformational dynamics can be obtained.

Otherwise, the manuscript is clearly and nicely written and I do not propose neither minor revision of the text nor revision of the results. This manuscript can be accepted as it is according to my best opinion.

Reviewer #2 (Remarks to the Author):

In this manuscript, Schmidt and coworkers reported the NMR study of a DNA duplex containing Hg-mediated T-C pairs. As concluded by the authors, the DNA adopts two different conformations in solution, with a molar ratio of 3:1. The results presented here provide structural evidences on the interaction of Hg with the bases, including (N3)T-Hg-(N3)C and (N3)T-Hg-(N4)C. Such interactions have been predicted or observed in other DNA duplex structure previously. Presence of different T-Hg-C interaction modes caused some conformational difference of the DNA duplexes,

however, as depicted in Figure 3 of the manuscript, the overall structures of the DNA duplexes are very similar. In the supplementary section, the author presented the spectra of the DNA duplex in the absence of Hg ion, however, the corresponding structure was not provided. Comparison of the structures in the presence or absence of Hg ion is important to clarify the conformational difference driven by the Hg ion. In the opinion of this referee the details added by this study are only incremental and the revealed flexibility of the Hg base pairs probably means that it is difficult to use this particular sequence in the rational design of DNA nanodevices, which normally undergo large conformational changes. This manuscript would be more suited to publication in a specialist journal such as nucleic acids research.

Reviewer #3 (Remarks to the Author):

This paper describes solution-phased NMR study on the dynamic changes in duplex DNA structures containing two different C-Hg(II)-T base pairs in equilibrium. The experiments were well executed and the explanation sounds good. However, I do not believe that this paper deserves publication in Nature Communications for the following reasons.

(1) The generality of this structural fluctuation is not clearly shown. The authors should examine T-Hg(II)-C base pairs in some other DNA duplexes with different sequences.

(2) What is the biological significance of this structural fluctuation?

(3) Why does the (N3)T-Hg-(N4)C base pair induce a more A-form-like duplex? The authors should provide a possible explanation or hypothesis even though it is difficult to prove.

(4) While the (N3)T-Hg-(N4)C pair is neutral, the (N3)T-Hg-(N3)C pair has a positive charge. The authors should discuss the influence of the charges on the equilibrium.

(5) Is it possible to change the ratio between two isomers by altering pH or ionic strength?

(6) The following papers should be cited in the introduction: (i) Kondo, J.; Tada, Y.; Dairaku, T.; Hattori, Y.; Saneyoshi, H.; Ono, A.; Tanaka, Y. Nat. Chem. 2017, 9, 956.

(ii) Takezawa, Y.; Müller, J.; Shionoya, M. Chem. Lett. 2017, 46, 622.

Reviewers' comments:

Reviewer #1 (Remarks to the Author):

Luedtke and co-workers reported in the manuscript Dynamic isomerization of a metallated double helix preparation and NMR measurements of novel metallo-DNA duplex including two C-Hg(II)-T base pairs separated by six normal Watson-Crick base pairs. This work represents outstanding contribution to the research of metallo-nucleic acids within the field of general metallo-organic chemistry.

We appreciate the reviewer's glowing opinion regarding the high importance of this manuscript.

The modification of nucleic acids has been recently recognized important namely in connection with development of functional molecules and advanced nano-materials. In that regard, the metallo-DNAs including "all-natural" nucleotides are particularly attractive because the platform enables relatively straightforward production of DNA molecules that can be functionalized upon adding of suitable metal. While the metal binding to several base pairs was recently determined only very little is known about conformational dynamics of metallo-DNAs. That is why I consider the work by Luedtke and co-workers significant for the field; new metal-binding within C-Hg(II)-T pair was characterized unambiguously and in detail, local conformational dynamics of the C-Hg(II)-T involving "major" and "minor" metal binding was determined inclusive respective rate constants, and particularly, the conformational dynamics of the tandem metallo-base pairs within metallo-DNA was determined

Astonishingly, the conformational dynamics of the two C-Hg(II)-T base pairs was concerted, so that only “major” or “minor” tandem occurs at once. Plausible validation of the long-distance coupling of the dynamical states within C-Hg(II)-T pairs mediated by nucleic acid backbone is unique even in the context of general structural biochemistry. Moreover, the concerted dynamics of metallo-base pairs induced concerted exchange of B and A classes of the backbone conformations. I do consider the discovery of coupled local dynamics within tandem C-Hg(II)-T with the global dynamics of whole metallo-DNA probably the most significant outcome of this work. It can have impact on progress within the field including future applications. Similar complex studies on metallo-nucleic acids are very scarce and I am not aware of any other that provides such a complex picture. This work demonstrated that dynamics of the 3D architecture of metallo-DNA is well-controllable by sequence programming of metallo-base pairs. The metallo-DNA thus can be used as template to study number of biochemical processes, to benchmark other spectroscopic measurements and to develop/calibrate force-fields for molecular dynamics simulations. May the authors consider to highlight this fact already in the title? (e.g.: Concerted dynamics of metallo-base pairs and whole metallo-DNA due to the metal isomerization (?))

We agree that our manuscript provides sufficiently detailed and precise data to develop/calibrate force-fields for molecular dynamics simulations. We appreciate the reviewer's suggestion for a new title. Together with the new experiments described below, we have changed the title of the manuscript to “Concerted dynamics of metallo-base pairs in an A/B-form helical transition”. We have removed the word “isomerization” since ionization of the ligand is also occurring.

I would like to address one issue concerning sustainability of the metal-binding/conformational dynamics due to pH. The NMR measurement was carried out at pH 7.8. When considering the Major and Minor metal bindings, the effect of pH may affect their dynamical exchange. Can the authors explain whether the effect on conformational dynamics due to different pH can be assumed? If yes, the pH-switch or pH-tuner of conformational dynamics can be obtained.

The impact of pH is certainly an important question. We have therefore conducted a new series of pH-dependent measurements. The results have revealed slower overall exchange rates with increasing pH, yet the ratios of the two species remained the same from pH 6 - 9. We have added these observations to the paper (Figures S32 – S35, SI). However, the only type of pH “switching” activity we observed was a loss of Hg^{II} from the duplex at pH < 6. Regarding the need for a more functionally useful, reversible switch, this issue was also been raised by reviewer #2, and we have addressed this by developing a DNA sequence capable of undergoing a reversible B → A-form transition upon addition/removal of Hg^{II}. This is presented in the last section of results in the revised manuscript.

Otherwise, the manuscript is clearly and nicely written and I do not propose neither minor revision of the text nor revision of the results. This manuscript can be accepted as it is according to my best opinion.

We thank the reviewer for their assessment.

Reviewer #2 (Remarks to the Author):

In this manuscript, Schmidit and coworkers reported the NMR study of a DNA duplex containing Hg-mediated T-C pairs. As concluded by the authors, the DNA adopts two different

conformations in solution, with a molar ratio of 3:1. The results presented here provide structural evidences on the interaction of Hg with the bases, including (N3)T-Hg-(N3)C and (N3)T-Hg-(N4)C. Such interactions have been predicted or observed in other DNA duplex structure previously.

We appreciate the reviewer's insightful and critical assessment of our manuscript. Their comments have served as an important catalyst for making significant improvements. That being said, prior to our study, there were only two (and directly conflicting) proposals for the structure of C-Hg-T base pairs. A preliminary proposal for (N3)T-Hg^{II}-(N3)C coordination was shown in a review paper in 2011 (*Chem. Soc. Rev.* 40, 5855–5866 (2011).) This proposal was based on structural homology with T-Hg^{II}-T, and the (very) small increases in thermal stabilities of duplexes containing C-T mismatches upon addition of Hg^{II}. In contrast, crystal screening of various oligonucleotides and metal ions produced an X-ray structure of a short (8-mer), A-form DNA sequence containing two C-Hg^{II}-T base pairs with a metal binding mode involving the exocyclic amine (N4) of a deprotonated cytosine "C" residue and (N3) of thymine "T" (*Nucleic Acids Res.* 45, 2910–2918 (2017)). However, the global A-form structure observed in this crystal structure was inconsistent with our own circular dichroism (CD) data of slightly longer, 14 – 21-mer duplexes containing one or two C-Hg-T base pairs (*Nucleic Acids Res.* 46, 6470–6479 (2018)). Our CD data suggested the presence of B-form helices, and little-to-no changes in their global conformation upon adding Hg^{II}. The exact metal binding mode(s) and global structural characteristics duplex DNA containing C-Hg^{II}-T base pairs in solution were therefore completely unclear prior to our extensive NMR studies that required both ¹⁵N-labeled DNA and ¹⁹⁹Hg enriched mercury salts to complete. We have modified the introduction to clarify this important fact.

Presence of different T-Hg-C interaction modes caused some conformational difference of the DNA duplexes, however, as depicted in Figure 3 of the manuscript, the overall structures of the DNA duplexes are very similar.

The first version of Figure 3 was indeed generated in such a way to maximize the overlap between the two structures. To make the significant differences between the two structures more apparent, we have aligned them according to the first three residues of each duplex in the revised version of the manuscript (now Figure 4B). This now allows the reader to see the greater axial bending and the deeper, narrower major groove present in the minor versus major Hg^{II}-bound duplexes.

In the supplementary section, the author presented the spectra of the DNA duplex in the absence of Hg ion, however, the corresponding structure was not provided. Comparison of the structures in the presence or absence of Hg ion is important to clarify the conformational difference driven by the Hg ion.

Indeed, previous studies have mostly focused on changes in structure upon small molecule binding. We have worked very hard to address this important point over the past five months. We have succeeded in generating the very first, high-resolution structure of a duplex DNA containing one or more C-Hg^{II}-T base pairs. We have added the structure of this metal-free duplex to the paper, and have fully integrated its analysis into comparisons with the two metal-bound structures. I hope the reviewer agrees that the interesting differences between the structures justify our extensive efforts. In particular, these results give insight into our design and identification of function of the first example of a metal ion-dependent global conformational change from B-form to A-form DNA.

In the opinion of this referee the details added by this study are only incremental and the revealed flexibility of the Hg base pairs probably means that it is difficult to use this particular sequence in the rational design of DNA nanodevices, which normally undergo large conformational changes.

Our results demonstrated that metal-nucleobase ionization and isomerization of the two C-Hg^{II}-T base pairs are directly coupled to each other over long distances (20 Å) via a global change in helical conformation. Furthermore, the central helical region located between the two metal centres exhibits much more A-form character as compared to the near-canonical B-form duplex between the two C-T mismatches of the metal-free duplex. These observations suggested that placing numerous C-T mismatches throughout a repetitive duplex sequence could cause a global B → A helical transition upon addition of Hg^{II}. To test this possibility, we prepared and analysed a small library of hairpin duplex DNAs (n = 10) and found one example that reversibly switched between an A-form and B-form helix upon addition/removal of Hg^{II}. In addition to its broad implications in structural biology and biochemistry, reversible A/B-form helical switching can be potentially utilized in the development of advanced DNA-based materials and devices. We have added another author to the paper, Ashkan Karimi, who conducted these important experiments.

This manuscript would be more suited to publication in a specialist journal such as nucleic acids research.

We respectfully disagree with this conclusion, especially in light of the new results added to our revised manuscript.

Reviewer #3 (Remarks to the Author):

This paper describes solution-phased NMR study on the dynamic changes in duplex DNA structures containing two different C-Hg(II)-T base pairs in equilibrium. The experiments were well executed and the explanation sounds good.

We thank the reviewer for their overall positive assessment of our manuscript.

However, I do not believe that this paper deserves publication in Nature Communications for the following reasons.

(1) The generality of this structural fluctuation is not clearly shown. The authors should examine T-Hg(II)-C base pairs in some other DNA duplexes with different sequences.

As observed in our “minor” duplex structure, an X-ray structure of a short DNA sequence (GCCCGTGC) containing two C-Hg^{II}-T base pairs (in red) exhibited (N3)T-Hg^{II}-(N4)C connectivity in the solid state (*Nucleic Acids Res.* 45, 2910–2918 (2017)). The global A-form structure observed in the crystal structure was inconsistent with our circular dichroism (CD) data collected in solution using five different, slightly longer (14 to 21-mer) duplexes containing one or two C-Hg-T base pairs (*Nucleic Acids Res.* 46, 6470–6479 (2018)). These CD spectra suggested B-form helices, and little-to-no changes in global conformation upon adding Hg^{II}. Given the observation that (N3)T-Hg^{II}-(N3)C binding present in our major structure is associated with more B-form character, and that the (N3)T-Hg^{II}-(N4)C present in the minor structure is associated with more A-form character, a general picture now emerges. (N3)T-Hg^{II}-

(N3)C binding and B-form structure is likely dominant in most DNA sequences, in direct analogy with T-Hg^{II}-T base pairs (*Angew. Chem. Int. Ed.* 53, 2385–2388 (2014)) as originally proposed (*Chem. Soc. Rev.* 40, 5855–5866 (2011)). However, the 10-fold larger ¹J coupling constants (¹⁵N, ¹⁹⁹Hg) observed for (N4)C-Hg^{II} (1052 Hz) connectivity in the minor duplex as compared to the (N3)C-Hg^{II} (114 Hz) in the major structure are indicative of stronger nucleobase-metal-nucleobase binding present in (N3)T-Hg^{II}-(N4)C of the minor structure, due to the ionization of the cytidine. The minor duplex, however, also exhibits a greater degree of structural perturbation away from a canonical B-form duplex that likely destabilizes the complex as compared to the major form. The lower bonding energy of (N3)T-Hg^{II}-(N3)C versus (N3)T-Hg^{II}-(N4)C is therefore compensated by the overall higher stability of B-form versus A-form in the major and minor structures, respectively. However, in certain situations that start to favour A-form (crystallization, dehydration, G/C-rich DNA sequences, etc.) the stronger (N3)T-Hg^{II}-(N4)C coordination mode starts to dominate. We were therefore very fortunate to choose a DNA sequence where both species could be observed, thereby allowing explanation of these disparate, previous observations. The preparation and analysis of additional, highly expensive, isotopically labelled DNA sequences using ¹⁵N NMR will not change this fact. The new data added to the paper (compare entries ODN¹² and ODN¹³, Table 2) further demonstrate a large impact of flanking sequences.

(2) What is the biological significance of this structural fluctuation?

In general, conversions between A-form and B-form structures are of fundamental importance in structural biology and biochemistry. They serve to regulate genome structure, protein and small molecule binding. Polymerases are known to mis-incorporate T across from T to give T-Hg^{II}-T base pairs in vitro. It is currently unknown, however if T-Hg^{II}-T or C-Hg^{II}-T are transiently formed in vivo. Given the new results in our paper regarding B → A-form transition upon addition of Hg^{II}, we have therefore focused potential applications of the current work towards materials and nanodevices.

(3) Why does the (N3)T-Hg-(N4)C base pair induce a more A-form-like duplex? The authors should provide a possible explanation or hypothesis even though it is difficult to prove.

The answer to this question is partially addressed above in response to comment #1 regarding the sequence context, A-form versus B-form propensities, and bonding strengths of the two binding modes. There is clearly a dynamic relationship between (N3)T-Hg-(N4)C connectivity and A-form propensity of the duplex, each making an energetic contribution. Regarding the exact structural details and causality of this relationship, it is indeed a very difficult to address these using NMR of highly dynamic, short duplex DNA. It is informative to again look at the highly precise X-ray structure of a short, A-form DNA sequence (GCCCGTGC) containing two C-Hg^{II}-T base pairs (in red) that exhibited (N3)T-Hg^{II}-(N4)C connectivity in the solid state (*Nucleic Acids Res.* 45, 2910–2918 (2017)). This paper suggests that the A-form structure is more accommodating towards of (N3)T-Hg-(N4)C connectivity due to the combined effects of geometry, shearing between the bases, neighbouring base effects, and a bridging water molecule in the minor groove. This is now mentioned in the discussion section.

(4) While the (N3)T-Hg-(N4)C pair is neutral, the (N3)T-Hg-(N3)C pair has a positive charge. The authors should discuss the influence of the charges on the equilibrium.

This is addressed in point (#1) above. We have accordingly modified the discussion section.

(5) Is it possible to change the ratio between two isomers by altering pH or ionic strength?

This is a good question. We conducted new experiments to evaluate the effects of altering pH and ionic strength and added these to the paper. To evaluate the impact of variable ionic strength, samples of the metallo duplex were prepared in the presence of 50 mM, 200 mM and 500 mM NaClO₄. The overall exchange rates of interconversion between the two structures decreased with increasing ionic strength, but the ratio of the two structures remained the same (Figures S27 – S31, SI). Likewise, pH-dependent measurements revealed slower overall exchange rates with increasing pH from pH 6 – 9 while maintaining the same ratio (Figures S32 – S35, SI). The lower rates of exchange correlate with the increasing persistence length and therefore rigidity of the duplex with increasing pH and ionic strength (*Phys. Rev. Lett.* (2019) 122, 028102; *Chin. Phys. Lett.* (2014) 31, 028701).

(6) The following papers should be cited in the introduction. (i) Kondo, J.; Tada, Y.; Dairaku, T.; Hattori, Y.; Saneyoshi, H.; Ono, A.; Tanaka, Y. *Nat. Chem.* 2017, 9, 956.
(ii) Takezawa, Y.; Müller, J.; Shionoya, M. *Chem. Lett.* 2017, 46, 622.

We have added these references to the introduction.

Reviewers' comments:

Reviewer #1 (Remarks to the Author):

Luedtke and co-workers considered all my comments and revised the manuscript with care and accordingly. The dependence of metal-induced dynamical behavior of Hg-DNA was solidly refined in a series of new experiments. The issue was further expanded and generalized also due to the comments by other reviewers.

Overall, the revision added to the value of this high-quality manuscript.

I recommend again the work for publishing in Nature Communications.

Dr. Vladimír Sychrovský

Czech Technical University, Technická 2, 166 27, Praha, Czech Republic,
Institute of Organic Chemistry and Biochemistry, Academy of Sciences of the Czech Republic,
Flemingovo náměstí 2, Praha, Czech Republic

Reviewer #2 (Remarks to the Author):

The authors have revised their manuscript to account for the comments of all reviewers. They made many changes to the manuscript, and these changes appear to address comprehensively all of the prior review comments. I acknowledge the extensive additional experiments, which have significantly improved the quality of the manuscript. With all these new data included, I find the revised version greatly improved and is acceptable for publication.

Reviewer #3 (Remarks to the Author):

The manuscript was substantially improved with several new data. In particular, Hg(II)-induced B-to-A transition presents a potential application of C-Hg-T pairing. These new results and revised discussion improved the novelty and importance of this research. Thus, I recommend this manuscript for publication while I still suggest a few more minor revisions as follows. (1) I would suggest the authors to conduct two more control experiments to confirm that the C-Hg-T pairing surely induced the B-to-A transition of ODN13. Specifically, the fluorescence anisotropy analysis and the CD measurement should be conducted with control DNA strands in which C-T pairs in ODN13 are replaced by T-T mismatches or matched base pairs (A-T etc.). (2) Although the authors added a new discussion on the impact of pH, they did not provide any direct answers to the comment that I sent before; "While the (N3)T-Hg-(N4)C pair is neutral, the (N3)T-Hg-(N3)C pair has a positive charge. The authors should discuss the influence of the charges on the equilibrium." I suggest the authors to discuss the influence of the different charges of the two base pairs more clearly.

Reviewer #3 (Remarks to the Author):

The manuscript was substantially improved with several new data. In particular, Hg(II)-induced B-to-A transition presents a potential application of C-Hg-T pairing. These new results and revised discussion improved the novelty and importance of this research. Thus, I recommend this manuscript for publication while I still suggest a few more minor revisions as follows. (1) I would suggest the authors to conduct two more control experiments to confirm that the C-Hg-T pairing surely induced the B-to-A transition of ODN13. Specifically, the

fluorescence anisotropy analysis and the CD measurement should be conducted with control DNA strands in which C–T pairs in ODN13 are replaced by T–T mismatches or matched base pairs (A–T etc.).

According to the reviewer's suggestion, we have taken the sequence of ODN13 and replaced the C–T mismatches with A–T base pairs to give ODN14. This hairpin (Fig. S43) was analysed for its ability to fold into a global A-form duplex using both neomycin binding (Table 2), as well as CD spectroscopy (Fig. S47i). According to the results from both experiments, hairpin ODN14 does not fold into a global A-form duplex upon addition of Hg^{II}. We have added these results to the manuscript and SI.

(2) Although the authors added a new discussion on the impact of pH, they did not provide any direct answers to the comment that I sent before; "While the (N3)T-Hg-(N4)C pair is neutral, the (N3)T-Hg-(N3)C pair has a positive charge. The authors should discuss the influence of the charges on the equilibrium." I suggest the authors to discuss the influence of the different charges of the two base pairs more clearly.

As presented in the previous revision, changing the pH or ionic strength had little or no significant impact on the position of the equilibrium. The different net changes of the two types of metal complexes therefore do not play a dominant role in the position of the equilibrium, and therefore, there is little to discuss about this point. A number of compensating effects must be present. To address the reviewer's insistence, we have introduced the following two sentences into the caption of the Figure 6 caption.

The Hg^{II}-ligand complexes present in the major species are net positive (+1), whereas those of the minor species are neutral. The combined contributions of more favorable electrostatic interactions and near-canonical B-form helical structure present in the major species likely outweigh the stronger ligand-metal-ligand binding interactions present in the minor duplex.